# QuEChERS LC–MS/MS Screening Method for Mycotoxin Detection in Cereal Products and Spices

**DOI:** 10.3390/ijerph18073774

**Published:** 2021-04-04

**Authors:** Licia Pantano, Ladislao La Scala, Francesco Olibrio, Francesco Giuseppe Galluzzo, Carmelo Bongiorno, Maria Drussilla Buscemi, Andrea Macaluso, Antonio Vella

**Affiliations:** 1Istituto Zooprofilattico Sperimentale della Sicilia, Via Gino Marinuzzi 3, 90129 Palermo, Italy; licia.pantano@izssicilia.it (L.P.); ladislao.lascala@gmail.com (L.L.S.); francesco.olibrio@gmail.com (F.O.); bongiornocarmelo81@gmail.com (C.B.); drussilla.buscemi@izssicilia.it (M.D.B.); amaca258@gmail.com (A.M.); laboratorio.residui@gmail.com (A.V.); 2Dipartimento di Scienze della Vita, Università degli studi di Modena e Reggio Emilia, Via Università 4, 41121 Modena, Italy

**Keywords:** mycotoxin, QuEChERS, LC–MS/MS, spices, cereals

## Abstract

We developed and validated a screening method for mycotoxin analysis in cereal products and spices. Ultra-high-performance liquid chromatography coupled with tandem mass spectrometry (UHPLC–MS/MS) was used for the analysis. Dispersive solid-phase extractions (d-SPEs) were used for the extraction of samples. Ochratoxin A (OTA), zearalenone (ZEA), aflatoxins (AFLA; AFB_1_, AFB_2_, AFG_1_, AFG_2_), deoxynivalenol (DON), fumonisin (FUMO; FB_1_, FB_2_, FB_3_), T2, and HT2 were validated in maize. AFLA and DON were validated in black pepper. The method satisfies the requirements of Commission Regulation (EC) no. 401/2006 and (EC) no. 1881/2006. The screening target concentration (STC) was under maximum permitted levels (MLs) for all mycotoxins validated. The method’s performance was assessed by two different proficiencies and tested with 100 real samples.

## 1. Introduction

Mycotoxins represent a group of secondary metabolites with different pharmacological and toxicological aspects [1]. Alkaloids of *Claviceps purpurea* such as ergotamine, ergometrine, and semi-synthesis derivate have been used in the therapy of Parkinson’s disease, post-partum hemorrhage, and hemicrania [2]. The toxicological effects of other mycotoxins, such as aflatoxins (AFLA; AFB_1_, AFB_2_, AFG_1_, AFG_2_), fumonisin (FUMO; FB_1_, FB_2_, FB_3_), deoxynivalenol (DON), T-2, and HT-2, have been well documented and studied in the literature [3,4,5,6].

Mycotoxicosis is a human and animal disease caused by ingestion, inhalation, or skin contact of mycotoxins [7]. The symptoms, target organ of systemic toxicity, and clinical outcome depend on several parameters such as type of mycotoxins, intake levels, time, and route of exposures [8]. Mycotoxicosis can be acute or chronic, and different symptoms characterize these two forms.

AFB_1_, AFB_2_, AFG_1_, and AFG_2_ are produced by fungi of the genus *Aspergillus*. The most representatives fungi that produce AFLA are *Aspergillus flavus* and *Aspergillus parasiticus* [9,10]. AFLA are characterized by a lipophilic structure (Figure 1) derived from the same precursor, versiconal hemiacetal acetate [11].

The acute ingestion of AFLA (aflatoxicosis) can lead to several symptoms that include gastrointestinal problems (diarrhea, abdominal pain), nervous system dysfunctions (epilepsy, coma), liver damage (jaundice, hepatitis), and even death [12,13]. Chronic exposure to AFLA is associated with multiple-organ cancer, immunosuppression, and other diseases [14]. AFB_1_, AFB_2_, AFG_1_, and AFG_2_ are classified as group 1 (carcinogenic to humans) by the Agency of Research on Cancer (IARC) and have mutagenic and teratogenic effects in humans [15]. Once ingested, AFLA are converted by cytochrome P450 into high reactive epoxides that can create adducts with nucleobases [16]. Hepatocellular carcinoma (HCC) is strictly correlated with dietary exposure to AFB_1_ and adducts excreted in urine [17,18].

FUMO (FB_1_, FB_2_, FB_3_) are produced by fungi of the genus *Fusarium* [19]. FB_1_ contamination is common in cereals, and it is the most toxic FUMO [20]. Acute ingestion of FUMO can cause gastrointestinal problems, and they are considered possibly carcinogenic to humans (group 2B) by IARC [15,21]. FUMO can interfere with folic acid metabolism (teratogenic effects), cause inhibition of sphingolipid biosynthesis, and have carcinogenic effects [11,21]. They are polar compounds and are not soluble in non-polar solvents (Figure 1) [21]. Chronic exposure to AFB_1_ and FUMO can lead to liver cancer (sum of carcinogenic effect) [22].

*Fusarium* species also produce DON, which is one of the most common mycotoxins in cereals [23]. It is considered not classifiable as to carcinogenicity to humans (group 3) [15]. The acute toxicity is mainly gastrointestinal, with nausea, diarrhea, and abdominal pain [24]. DON is also called vomitoxin since it can induce emesis [25]. It can also cause dysfunctions of the immune, neuroendocrine, and cardiovascular systems [26]. DON is a polar molecule that can resist at high temperatures, and it is soluble in polar organic solvents [27,28]. It is classified as non-macrocyclic trichothecenes [29].

Non-macrocyclic trichothecenes also include T2 and HT2 (C-4 deacetylated form of T2, Figure 1) produced from *Fusarium* species [30]. The name derived from trichothecin, the first non-macrocyclic trichothecene isolated in 1948 from *Trichothecium roesum* [11]. T2 is the most toxic among all trichothecene [31]. T2 and HT2 have been reported frequently in cereal-based products [32,33]. Acute toxicity symptoms are similar to DON [34]. T2 can inhibit DNA, RNA, and protein synthesis [35]; can induce apoptosis; and has immunotoxic effects [32]. T2 and HT2 can resist temperature, and they are deactivated by low or high pH [35]. 

Ochratoxin A (OTA) is the most important and toxic mycotoxin among ochratoxins [36]. It is an isocumaric derivate with a β-phenylalanine (Figure 1) [11]. *Aspergillus* and *Penicillium* species can produce OTA; *Aspergillus ochraceus* and *Penicillium verrucosum* are the most common [37]. It is located in group 2B in the IARC classification [15]. Its toxicological activity can affect neuronal, renal, and immune systems [36]. OTA has a high affinity for serum albumin and, therefore, it is characterized by a high plasma half-life [38]. It has been found in several foods such as milk, coffee, wine, and vegetables [38]. It also has been reported in baby food [39]. OTA is soluble in polar organic solvents [40].

Zearalenone (ZEA) is a resorcylic acid lactone (Figure 1) produced by the *Fusarium* genus species [8]. It is classified as group 3 by IARC [15]. It is associated with endocrine alteration with clinical manifestations of hyperestrogenism [41]. Swine are the most sensitive species to the toxic effects of ZEA; clinical signs of acute mycotoxicosis are swelling of the vulva, prolonged estrus, pseudopregnancy, and infertility [37]. ZEA can be found in crops, wheat, maize, and snacks [42]. 

Mycotoxins in food are unavoidable contaminants [13]. High temperatures and bad storage conditions can increase mycotoxin contamination in food [43,44,45,46]. They can resist standard cooking methods [47,48,49], leading to serious health concerns [33].

In developing countries, there have been relatively recent episodes of mycotoxicosis. One-hundred and twenty-five people died in Kenya (Africa) in 2004 [50], and there were 100 victims in 1974 in India [51]. There were alimentary toxic aleukia (ATA) cases in the USSR in the 1950s associated with consuming trichothecenes in wheat [37]. 

Cereals such as maize, oat, and wheat are considered the most susceptible to mycotoxin contamination [52,53,54,55,56,57,58,59]. It was proven that 25% of cereals consumed in the world in 1988 were contaminated by mycotoxins [60]. Products derived from cereals such as pasta, bread, and snacks can contain cereal-derived mycotoxins [61,62].

One of the biggest problems related to the cereal contaminations of mycotoxin is the carry-over effect. Mycotoxins consumed in feeding stuff are absorbed and metabolized by animals and can be found in meat, eggs, and milk [63].

Different cereal-based products for human consumptions are mixed with black pepper for flavoring purposes or preservatives, leading to possible increases in mycotoxin contents [60]. Spices are usually dried on the ground in the open air in poor hygienic conditions that promote the growth of molds and production of mycotoxins [64,65]. The Rapid Alert System for Food and Feed (RASFF) reported that the third most common matrices with AFLA in Europe are herbs and spices [66]. 

The Commission Regulation (EC) no. 401/2006 established MLs of mycotoxins for different food and feed [67]. AFLA (AFB_1_, the sum of AFB_1_ + AFB_2_ + AFG_1_ + AFG_2_, AFM_1_), OTA, patulin, DON, ZEA, FUMO (FB_1_ + FB_2_), T-2 and HT-2, citrinin, ergot sclerotia, and ergot alkaloids were regulated in Europe. The regulation does not consider the assessment of exposure to the sum of mycotoxins [33] with potentially higher exposure. Furthermore, the presence of masked mycotoxins can lead to an underestimation of their concentration in a sample [68].

There are several analytical techniques to analyze mycotoxin in different matrices [69,70,71,72,73]. HPLC methods associated with fluorescence detection (FLD) or MS/MS are the most used for mycotoxin analysis [74,75,76,77,78,79,80,81]. In most cases, methods are validated on a single matrix and do not meet the supervisory bodies’ needs that require different mycotoxin limits for different matrices [67,77,78,82]. Regarding extraction method, solid-phase extraction (SPE), solid–liquid extraction (SLE), and liquid–liquid extraction (LLE) are common techniques used for LC–MS/MS analysis and also for mycotoxin [74,82]. The challenge of mycotoxin analyses is that they have different proprieties and polarity [83]. Therefore the right choice of the extraction method can be difficult [84].

The QuEChERS (Quick Easy Cheap Effective Rugged and Safe) method is becoming one of the most used dispersive solid-phase extraction (d-SPE) methods in food safety [69,85,86,87,88]. According to Web of Science, more than 1200 scientific papers cover aspects or use QuEChERS for extraction procedures. The common steps of QuEChERS methods are LLE between an organic and an inorganic phase, addition of salts, agitation of the sample, removal of the supernatant, and d-SPE between supernatant and sorbents [89]. Competent authorities already use QuEChERS extraction to analyze various molecules such as pesticides and antibiotics [90,91]. 

Multiclass multi-residue methods (MRMs) with QuEChERS d-SPE are preferred over other cleanup solutions requiring numerous steps and cost more time. Fast cleanup reduces the costs, time, and space required with solid-phase cartridges [92]. 

QuEChERS does not require high solvents volume and can be performed with basic laboratory devices [93] and with an extraction procedure that allows for the analysis of 20–30 samples in 1 h.

The majority of methods are focused on increasing mycotoxins analyzed in a single matrix [94,95,96,97], especially in maize and in wheat [98].There are already QuEChERS mycotoxin procedures in the literature [99,100,101]. However, they are validated in a single matrix or use high-resolution mass spectrometry (HRMS), which is more expensive than an LC–MS/MS.

The present work aimed to validate QuEChERS extraction together with a fast and reliable LC–MS/MS method to detect 12 mycotoxins in cereals and 5 mycotoxins in black pepper, following the (EC) Regulation limits 1881/2006 and (EC) Regulation 401/2006.

## 2. Materials and Methods

### 2.1. Chemical and Standards

Methanol, acetonitrile, and formic acid were LC–MS-grades (>99.9%), and were purchased from Sigma-Aldrich (Amsterdam, Holland). Ultrapure water was obtained in the laboratory using a Milli-Q system (Millipore Burlington, MA, USA). OTA, ZEA, DON, AFB_1_, AFB_2_, AFG_1_, AFG_2_, FB_1_, FB_2_, FB_3_, T2, and HT2 were purchased by Sigma-Aldrich.

### 2.2. Materials

The products Supel QuE Citrate (EN) Tube (55227-U) and Supel QuE PSA (primary, secondary amine, EN) Tube (55228-U) were purchased from Sigma-Aldrich (Amsterdam Holland). The composition of 55227-U is 4 g MgSO_4_, 1 g NaCl, 0.5 g sodium citrate dibasic sesquihydrate, and 1 g sodium citrate tribasic dihydrate. 55228-U contains 0.9 g MgSO_4_ and 150 mg of Supelclean PSA.

### 2.3. Working Solutions

Standards were mixed to obtain the following working solution: OTA, AFLA (AFB_1_, AFB_2_, AFG_1_, AFG_2_), FUMO (FB_1_, FB_2_, FB_3_), ZEA, DON, MIX T2 (T2, HT2). Acetonitrile was used as solvent, except for SMix1 (low level) and Smix2 (high level), where methanol was used as solvent. A working solution was used to fortify the blank matrix sample, and low level and high level were used as calibration solutions (Table 1). 

### 2.4. Sample Preparation

The method was applied to screen mycotoxins in Sicily (Southern Italy). About 10 kg of maize and 5 kg of black pepper were collected from 2 different local vendors in Palermo (Sicily) and used for validation procedures. Samples were grounded using a Mixer B-400 laboratory mill by BÜCHI (Cornaredo, Italy) at ambient temperature with knives’ rotation speed of 9000 rpm. Samples grounded were stored at −10 °C until analysis.

### 2.5. Sample Extraction

About 5.0 ± 0.1 g of the sample was weighted in a falcon tube of 50 mL. A total of 150 μL of OTA-d5 with a concentration of 100 μg/L was added to all samples (3.0 μg/Kg, 7.5 μg/L). One sample for each matrix was fortified with the working solution as described above (Table 1). After 10 minutes, 10 mL of bidistilled water and 10 mL of an acetonitrile/formic acid solution (80:20 v/v) were added to the sample. The sample was vortexed for 15 min and left to rest for 15 minutes at −20 °C. A mixture of salt (4 g MgSO_4_, 1 g NaCl, 0.5 g sodium citrate dibasic sesquihydrate, 1 g sodium citrate tribasic dihydrate) was added, handle shacking occurred for about 1 minute, and the mixture was centrifugated for 10 min at 5000 rpm. The supernatant was transferred into a mix of salt, 900 mg MgSO_4_, and 150 mg Supelclean PSA. After 1 minute of handle shacking and 5 minutes of centrifugationat 5000 rpm, 3 mL of the supernatant was evaporated (40 °C) and dissolved in 600 μL of methanol/water (50/50 v/v). The sample was ready for the injection.

### 2.6. Instrumentation

The analysis was performed on a Thermo Fischer Ultra High Performance Liquid Chromatography(UHPLC ) system (Thermo Fisher Scientific, California, CA, USA) consisting of an ACCELA 1250 quaternary pump and an ACCELA autosampler. A Thermo Scientific Hypersil Gold reversed-phase UHPLC column (50 mm, 2.1 mm ID, 1.9 μm) was used for mycotoxin analysis.

The mobile phase (Table 2) was a time-programmed gradient using water (eluent A) and methanol (eluent B). Both contained 2.50 mM of ammonium formate and 0.1% formic acid. The chromatographic run started with 100% of A with a variation of 20% in 0.5 min. The conditions were maintained for 1 min, and then A decreased until a percentage of 40% in 0.1 min. Linear decrease of A occurred with a total percentage of B of 100% in 2.6 min. Conditions were maintained for 0.7 min, and the system returned to 100% A and 0% of B in 0.1 min for 1 min. 

A triple quadrupole TSQ Vantage (Thermo Fisher Scientific, California, CA, USA) in positive electrospray ionization (ESI) mode was used as a spectrometer. The product ion scans were obtained by a direct infusion of each analyte dissolved in methanol/water (50/50 v/v).

The ESI parameters were set as follows: capillary temperature 310 °C, vaporizer temperature 300 °C, sheath gas pressure 40 psi, auxiliary gas pressure 30 psi, capillary voltage 4.8 kV. Collision gas, peak resolution, scan time, and scan width parameters were set as described in [102]. Trace Finder version 4.1 from Thermo Fisher (Kandel, Germany) was used to record and elaborate data. Results obtained from the analyte’s direct infusion (parent, product 1, product 2, CE) and chromatography runs can be seen in Table 3.

### 2.7. Validation Procedure

The method was validated according to the EU Commission Decision 2002/657/EC, following the Council Directive 96/23/EC and Regulation (EC) no. 401/2006. Linearity, specificity, precision (repeatability and reproducibility within-laboratory), and ruggedness were determined. The specificity was determined by analyzing 20 blank and fortified samples for each matrix to assay the absence of interfering peaks. The linearity was tested with a standard curve of 5 points, including zero, as follows: AFB_2_- AFG_2_ (200–2000 μg/L), AFB_1_-AFG_2_ (800–8000 μg/L), DON (50,000–500,000 μg/L), FUMO (200,000–2,000,000 μg/L), T2-HT2 (125,00–125,000 μg/L), ZEA (37,500–375,000 μg/L), OTA (1500–15,000 μg/L). The linear coefficients for each calibration in curve were *r^2^* > 0.99. The precision was assessed by fortifying 20 samples at screening target concentration (STC) and by analyzing 4 of them each day for 5 days to calculate intermediate precision (RSDRi); then, the cut-off was calculated. As required from Commission Regulation (EC) no. 401/2006, STC must be under or equal with MLs report into (EC) Regulation 1881/2006. 

The ruggedness test was performed according to Youden [103] by determination of the effect of changing conditions (speed and time of centrifugation, time and speed of stirring, evaporation temperature). The identification of analytes was made by comparing the retention time in the sample (TR_s_) and the spiked sample (TR_a_) with a range of ±0.1 min. 

The semi-quantification of analytes was made by extrapolation of data obtained in the linear regression between low level and high level (Table 1), and the concentration in the sample was obtained with the following formula:C_C_ = C_S_ · D,(1)

C_c_ is the concentration in the matrix (μg/Kg), C_s_ is solution (μg/L), and D is the dilution factor.

Matrix effect (MEs) in maize and spice were calculated as described by Juan Sun et al. [99]:MEs = 100 (1 − (A_bm_/A_s_)) (2)

A_bm_ is the area in the blank matrix and A_s_ is the area of mycotoxin standard in solvent (n = 3).

Accuracy was evaluated with the extraction of 20 fortified samples for each matrix. 

### 2.8. Real Samples

After validation, the method was tested with real samples collected during an inspection in Palermo (Sicily). Maize (*n* = 25) and wheat (*n* = 25) were collected from the same local vendors; black pepper (*n* = 25) and coffee (*n* = 25) were from two different local supermarkets in Palermo (Sicily). All samples were transported daily in the lab and, if needed, grounded as described in Section 2.4. The extractions were performed as described in Section 2.5.

## 3. Results and Discussion

### 3.1. Method Development

#### 3.1.1. Extraction Solvent and Cleanup

The extraction method developed allows for extracting 12 mycotoxins in cereals and 5 in black pepper with a cheap and fast procedure. The best performance was attained using water and a mixture of acetonitrile/formic acid (80:20 v/v) Other ratios (60:40, 40:60, 20:80, 50:50) were examined to reach this conclusion (Figure 2). Acetonitrile and formic acid enhance analytical performance, as previously reported in the literature [104]. Acetonitrile/water extraction (in different percentages) is one of the most common mixtures used for mycotoxin analysis [105]. Acetonitrile can reduce the extraction of lipophilic materials such as fats and has a high capacity to extract molecules characterized by different polarities [106]. All mycotoxins analyzed are soluble in acetonitrile, and a higher percentage of acetonitrile can improve analytes’ extraction. OTA, AFLA, and ZEA are soluble in polar organic solvents such as methanol and acetonitrile [40,107,108]. FUMO are hydrophilic mycotoxins (Figure 1) and are soluble in the same solvents and water [109].

Regarding trichothecenes, T2, HT2, and DON are low soluble in water but highly soluble in ethanol and organic solvents [110]. For these reasons, acetonitrile, methanol, and water were used for the extraction procedure. Furthermore, water is added in high starch or low water matrices to reduce the interaction between them and analytes [111]. 

Formic acid and citrate salts decrease pH and contrast PSA’s effect of increasing pH in the second step [112,113]. MgSO_4_ and NaCl increase the recovery of polar analytes, and MSO_4_ with PSA performs better and increases mycotoxins’ recovery. [106,114].

The ratio between MgSO_4_ and PSA in the sorbent d-SPE step is always greater than 1 in all the literature, even if the quantity can change [106]. However, 900 MgSO_4_ and 150 mg of Supelclean PSA is already used in different analytical techniques that use QuEChERS methods for the extraction of analytes such as pesticides [115,116], likewise for the 4:1 w/w ratio between MgSO_4_ and NaCl as salt added in the extraction procedure [69,113,117,118]. 

#### 3.1.2. Matrix Effect in Mycotoxin Analysis

LC–MS/MS is susceptible to MEs, a common and unpredictable problem that can influence the validation process [119]. The ESI ionization is subject to ion suppression more than other atmospheric pressure ionization (API) techniques such as atmospheric pressure chemical ionization (APCI) [120].

Isotope-labeled internal standards can be useful to handle with MEs and are considered the “gold standard” approach [121]. This approach is useful for predicting the strong MEs in the analytical signal [122]. However, isotope-labeled internal standards are very expensive and do not exist for all mycotoxins, and recovery is not required in the screening method [123]. For these reasons, the use of a fortified sample for each analyte session seems more reasonable. Nevertheless, OTA-d5 was used in validation procedure and in real samples to evaluate the entire process. 

In the present work, a strong ME was observed for AFLA with a range of (+43.74, −20.18) in maize and (−26.17, −0.89) in black pepper (Table 4).

MEs have been reported several times in the literature, but with discordant data, enhancing or suppressing black pepper [122,124,125] and maize [122]. FUMO was strongly enhanced with an overall increase of 48.51%, as already reported by Beltrán et al. (2009) [104]. Curiously, OTA suppression in black pepper and maize was similar and had been reported before for maize [126].

The strong MEs in the mycotoxin analysis was caused by the lipid/water/protein content of matrices analyzed [127]. For these reasons, the analysis of different matrices should be validated according to (EC) no. 401/2006 that divides food into commodity groups to validate screening methods [67]. In this case, the research validated the commodity group of difficult or unique commodities (black pepper) that include cocoa beans and products thereof; copra and products thereof; coffee, tea, and licorice and the commodity group cereal grain and products thereof (maize), which include wheat, rye, barley, rice, oats, wholemeal bread, white bread, crackers, breakfast cereals, and pasta. Confirmatory methods must be validated in each matrix and for these reason screening methods, increasing the chance to discover new incidents and protect the consumers from high mycotoxin exposure [67].

### 3.2. Method Optimization

#### 3.2.1. Validation Parameters

The analytical parameters of the methods used for mycotoxin analysis are regulated in Europe by the Commission Regulation (EC) no. 401/2006 that defines the criteria of analysis for the official control of the levels of mycotoxins in foodstuffs [128]. The analysis was performed under RSDRi, and all cut-off levels were under STC. The linear correlation of level tested in the range of linearity was acceptable (*r^2^* > 0.99).

Regarding black pepper, maximum levels of OTA (15 μg/kg), AFB_1_ (5 μg/kg), and the sum of AFB_1_ + AFB_2_ + AFG_1_ + AFG_2_ (10 μg/kg) were established from current legislation [67]. More mycotoxins are regulated for maize and unprocessed cereals intended for direct human consumption: OTA (3 μg/kg), DON (750 μg/kg), ZEA (75 μg/kg), AFB_1_ (2 μg/kg), the sum of AFB_1_ + AFB_2_ + AFG_1_ + AFG_2_ (4 μg/kg), FB_1_ + FB_2_ (400 μg/kg), and the sum of T2-HT2 (3 μg/kg). All mycotoxin listed were validated according to (EC) no. 401/2006 regarding the screening method for mycotoxin analysis; the cut-off level must be equal or lower than the STC level, and the method developed was complied with. The validation data can be seen in Table 4.

#### 3.2.2. Instrumental Method

A cheap and simple screening method of 12 (cereal) and 5 (black pepper) mycotoxins was validated. Ammonium formate and formic acid were used to form ammonium adduct and protonated precursor ion, respectively. The ammonium adduct was selected as precursor ion only for T2 and HT2 (Table 3). All mycotoxins were detected within 4 min. The adequate resolution was obtained between ions with the same *m/z*, which would be indistinguishable from the mass spectrometer if they coalesced. LC–MS/MS is used frequently to analyze molecules regulated by EU Legislation [129,130,131,132], and it is possible to perform semi-quantitative analysis [133]. For this reason, LC–MS/MS was preferred over other screening analytical techniques such as enzyme-linked immunosorbent assay (ELISA) [134,135].

Several parameters can influence the performance of mycotoxin analysis. Regarding instrument setting, the positive ion mode (ESI^+^) was chosen because there is a better response for the overall of mycotoxin [104,127], especially AFLA that among mycotoxins are more regulated in the EU Legislation [67].

Water and methanol as mobile phases provided the best performance for peak resolution and run time for the chromatographic run. The same result was reported in the literature [84,99,127]. Ammonium acetate and formic acid addiction in the mobile phase increase the analytical performance [94,104,136,137]. The best results were achieved with 0.1% of formic acid and 2.5 mM of NH_4_COOH, as already reported by other authors [123].

With a total chromatography run of 6 min and an extraction procedure that takes approximately 1 hour, the method developed is faster than other methods already reported in the literature [138,139,140,141,142] and is useful for quick screening.

AFLA are not sensitive to heat treatment and can increase during the food storage period. A quick screening before storage can be useful to have some data on mycotoxins’ presence [48,143,144].

Black pepper is a less studied matrix, and increased data on mycotoxins presence can help in the risk assessment of mycotoxins exposure. This is especially the case because there is a possible co-occurrence of mycotoxin due to the multiple fungal infections [145] and because they are used as flavor-enhancers in convenience foods. 

However, this result has not been reached without compromises. Masked mycotoxins such as 3-acetyl deoxynivalenol (3-AcDON) and 15-acetyl-deoxynivalenol (15-AcDON) that present different toxicities [146] are not currently regulated in EU Legislation and were not analyzed. Some peaks are moderately overlapped, such as for FB_2_ and FB_3_; however (EC) no. 1881/2006 requires only sum of FB_1_ + FB_2_ (Figure 3). Ergot sclerotia and ergot alkaloids required from (EC) no. 1881/2006 were not analyzed.

### 3.3. Ring Test and Application to Real Samples

#### 3.3.1. Ring Test

The laboratories’ performance was assessed by proficiency tests (PTs) following ISO/IEC 17025:2018. The analytical quality of the validated method was assured by the participation in the interlaboratory study. The following PTs were purchased by Progetto Trieste (Test Veritas, Padova, Italy): MA2050 that consist of maize with an assigned values of AFB_1_+, AFB_2_+, AFG_1_+, AFG_2_ of 16.57 μg/kg expressed as sum; maize MA2051 with DON 702.14 μg/kg; ZEA 232.61 μg/kg; F2061 feed with 11.42 μg/kg AFB_1_; 2.42 μg/kg AFB_2_; 7.65 μg/kg AFG_1_; wheat WH2062 with DON 527.05 μg/kg and T2 18.31 μg/kg; dried figs DF2064 with AFB_1_ 7.30 μg/kg; AFG_1_ 5.35 μg/kg; and OTA 8.79 μg/kg; and GC + C2053 that consists of coffee with an assigned value of 7.53 μg/kg for OTA (Progetto Trieste, Test Veritas, Padova, Italy). The results of all tests were compliant with ISO/IEC 17025:2018. It is worth noting that according to (EC) no. 401/2006, dried figs are classified as “high sugar and low water content”, which is a commodity group not validated, and despite this, the result was compliant with ISO/IEC 17025:2018. 

#### 3.3.2. Application of the Method to Real Samples

All samples were compliant and following (EC) no. 1881/2006. One sample of maize resulted with OTA at 2.53 μg/Kg, and one sample of black pepper resulted with 1.85 μg/Kg of OTA and the contemporary presence of 0.358 μg/Kg of AFB_2_ (Table 5).

## 4. Conclusions

A new method for detecting 12 mycotoxins in cereals and 5 mycotoxins in spices (black pepper) was developed and validated according to (EC) no. 401/2006. QuEChERS extraction was used effectively. The best performances were obtained with acetonitrile/formic acid (80:20 v/v) as extraction solvent. Strong MEs were observed in all the FUMO analyzed in maize, while AFLA had enhancing or suppressing effects. In black pepper, there was a suppression of signals for all mycotoxins analyzed. Six PTs were developed to evaluate the performance of the method. The method was applied to 100 real samples (25 maize, 25 wheat, 25 black pepper, and 25 coffee). Two samples had a detectable amount of mycotoxin, maize (OTA, 2.53 μg/Kg), and black pepper (OTA, 1.85 μg/Kg, and AFB_2_, 0.358 μg/Kg). The method proposed is suitable for screening and routine analysis to monitor mycotoxins content in foodstuff following European Regulamentation. Further studies are needed to increase the number of mycotoxins analyzed and increase commodity groups analyzed with the same method.

## Figures and Tables

**Figure 1 ijerph-18-03774-f001:**
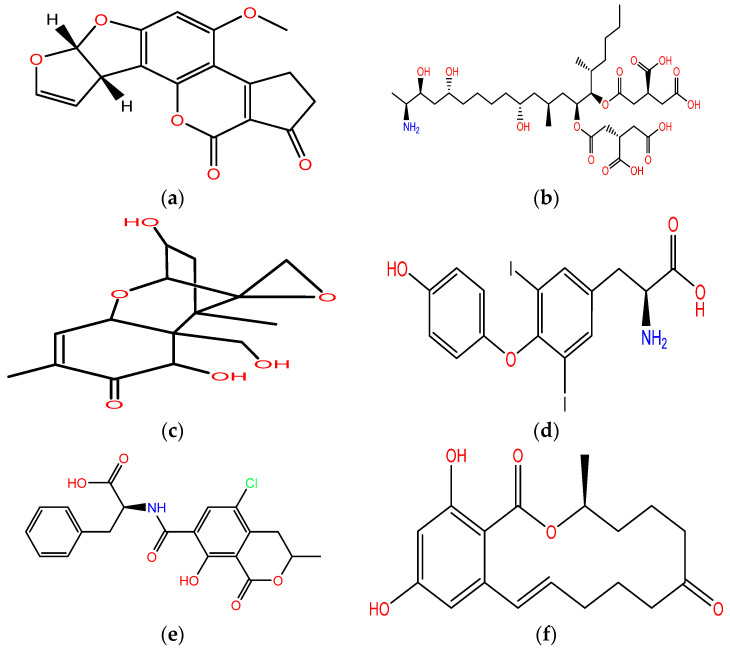
Molecular structure of some mycotoxins: (**a**) AFB_1_; (**b**) FB_1_; (**c**) deoxynivalenol (DON); (**d**) T2; (**e**) ochratoxin A (OTA); (**f**) zearalenone (ZEA).

**Figure 2 ijerph-18-03774-f002:**
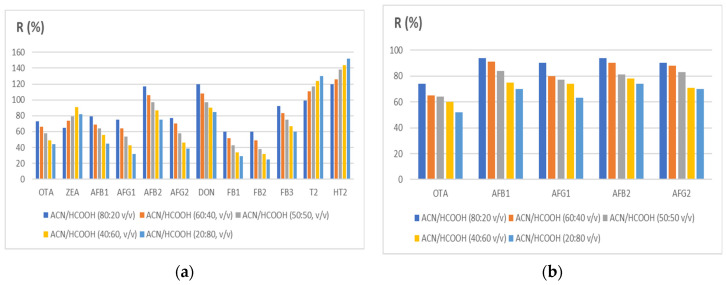
Effect of solvents on the extraction recovery: (**a**) maize; (**b**) black pepper. ACN, acetonitrile; HCOOH, formic acid. The increase of ACN increased the recovery of all mycotoxins.

**Figure 3 ijerph-18-03774-f003:**
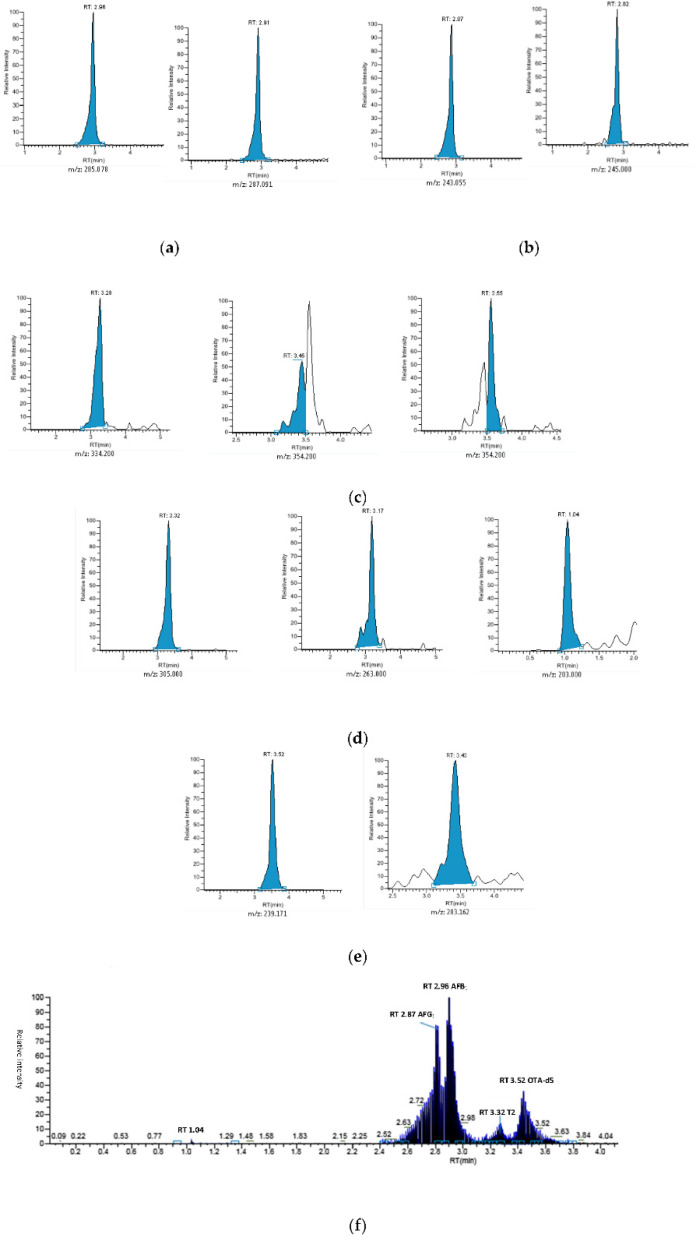
Ultra-high-performance liquid chromatography coupled with tandem mass spectrometry (UHPLC–MS/MS) chromatograms obtained from a blank maize spiked with working solution (Table 1). Fumonisins were spiked at 100 μg/kg for a better visualization. From left to right: (**a**) AFB_1_, AFB_2_; (**b**) AFG_1_, AFG_2_; (**c**) FB_1_, FB_2_, FB_3_; (**d**)T2, HT2, DON; (**e**) OTA, ZEA; (**f**) total ion chromatogram (TIC).

**Table 1 ijerph-18-03774-t001:** Working solutions were obtained by mixing standard solutions. Conc: concentration, BMF: blank matrix fortified, Vol: volume (μL).

Working Solutions		Conc.	Fortified Sample		SMix 1 (Low Level)		SMix 2 (High Level)	
			Conc μg/Kg (μg/L)	Vol ^2^ (μL)	Conc.	Vol ^3^ (μL)	μg/L	Vol ^3^ (μL)
OTA	OTA	100	3.0 (7.5)	150	1.5	15	7.5	75
AFLA	AFB_1_	100	1.6 (4.0)		0.8		4	
	AFG_1_	100	1.6 (4.0)	80	0.8	8	4	40
	AFB_2_	25	0.4 (1.0)		0.2		1	
	AFG_2_	25	0.4 (1.0)		0.2		1	
ZEA	ZEA	1000	75 (187.5)	375	37.5	37.5	187.5	187.5
FUMO	FB_1_	10,000	400 (1000)		200			
	FB_2_	10,000	400 (1000)	200	200	20	1000	100
	FB_3_	10,000	400 (1000)		200			
DON	DON	10,000	100 (250)	50	50	5	250	25
MIX T2	T2	1000	25 (62.5)		12.5			
				125		12.5	62.5	62.5
	HT2	1000	25 (62.5)		12.5			
OTA-d5 ^1^	OTA D5	100	3.0 (7.5)	150	7.5	75	7.5	75

^1^ Internal standard. ^2^ Volume of working solution to add to the fortified samples. ^3^ Volume of working solution added to obtain 1 mL of SMix 1 (low level) and 1 mL of SMix2 (high level).

**Table 2 ijerph-18-03774-t002:** The mobile phases in the chromatography run.

Time (min)	A (%)	B (%)
0	100	0
0.5	80	20
1.5	80	20
1.6	40	60
4.2	0	100
4.9	0	100
5	100	0
6	100	0

**Table 3 ijerph-18-03774-t003:** Retention time, the most abundant *m/z* ions and optimal collision energy (CE).

Mycotoxin	Rt	Parent	Product 1 (m/z)	CE (V)	Product 2 (m/z)	CE (V)
OTA	3.52	404.2 [M + H]^+^	239.2	27	221.7	37
ZEA	3.43	319.1 [M + H]^+^	283.2	20	187.0	22
AFB_1_	2.96	313.1 [M + H]^+^	285.1	25	241.0	38
AFB_2_	2.91	315.1 [M + H]^+^	287.1	27	259.0	30
AFG_1_	2.87	329.1 [M + H]^+^	243.0	27	311.0	25
AFG_2_	2.82	331.1 [M + H]^+^	245.0	30	313.0	30
DON	1.04	297.2 [M + H]^+^	203	20	249.2	15
FB_1_	3.28	722.2 [M + H]^+^	334.2	40	252.2	30
FB_2_	3.45	706.3 [M + H]^+^	354.2	37	336.1	37
FB_3_	3.55	706.3 [M + H]^+^	354.2	37	336.1	37
T2	3.32	484.2 [M + NH_4_]^+^	305.0	15	215.0	15
HT2	3.17	442.2 [M + NH_4_]^+^	263.0	15	215.0	15

**Table 4 ijerph-18-03774-t004:** Results of the validation procedure.

Sample Type	Mycotoxin	Linearity (μg/L)	Matrix Effect (%)	STC μg/kg	Cut-Off μg/kg	Repeatability	Recovery (%)
Maize	OTA	1.5–15	−25.40	3.0	0.93	1.81	73
	ZEA	37.5–375	−2.85	75	43	7.83	65
	AFB_1_	0.8–8	−12.18	1.6	0.95	0.95	79
	AFG_1_	0.8–8	+43.74	1.6	1.06	0.19	75
	AFB_2_	0.20–2	−20.18	0.4	0.37	0.14	117
	AFG_2_	0.20–2	+6.69	0.4	0.27	0.057	77
	DON	50–500	+37.06	100	75	36.7	120
	FB_1_	200–2000	+57.26	400	75	36.7	60
	FB_2_	200–2000	+66.82	400	46	141	60
	FB_3_	200–2000	+21.47	400	212	223	92
	T2	12.50–125	−13.35	25	23	1.99	99
	HT2	12.50–125	−2.21	25	26	6.02	120
Black pepper	OTA	1.5–15	−26.03	3.0	1.33	1.27	74
	AFB_1_	0.8–8	−13.64	1.6	1.44	0.09	94
	AFG_1_	0.8–8	−1.41	1.6	1.38	0.1	90
	AFB_2_	0.2–2	−26.17	0.4	0.35	0.03	94
	AFG_2_	0.2–2	−0.89	0.4	0.34	0.026	90

**Table 5 ijerph-18-03774-t005:** Results of the analyses on real samples. A total of 25 samples were analyzed for each matrix. Dash indicates that all results were under cut-off levels.

Sample Commodity	Detected Mycotoxin	Number of Sample with a Detectable Amount of Mycotoxin	Amount
Maize	OTA	1	2.53 μg/kg
Wheat	-	0	-
Black pepper	OTA AFB_2_	1 1	1.85 μg/kg 0.358 μg/kg
Coffee	-	0	-

## Data Availability

Not applicable

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
