# Peer review of "QuEChERS LC–MS/MS Screening Method for Mycotoxin Detection in Cereal Products and Spices"

_ijerph, 2021, doi:10.3390/ijerph18073774_

Round 1

Reviewer 1 Report

This research described a screening method for mycotoxin detection in cereal products and spices. Although the method might be effective for the selected mycotoxins, the obtained analysis results were not very advantageous, even if it was emphasized to be a QuEChERS method. It was not recommended for publication.

There are so many strategies for mycotoxins analysis, the author was suggested to offer one detailed comparison on the results. This may be effective to show the merits of this research.

Figure 1 was unclear, please clarify them.

Please unify the units in the article. It will help the readers to learn the article.

This research described a screening method for mycotoxin detection in cereal products and spices. Although the method might be effective for the selected mycotoxins, the obtained analysis results were not very advantageous. As a result, It was not recommended for publication.

In introduction part, it is recommended to provide more detailed information about the hazards of mycotoxins measured. The molecular structure of the mycotoxins tested should be provided, and more analysis of experimental results should be made accordingly.

Although the author proposed that the established method is a QuEChERS method, the above characteristics are not seen from the results presented. Please add relevant research data such as detection time, accuracy, etc. in the manuscript.

The chromatograms of the various substances provided are not clear, and the key information is missing. It is necessary to provide a complete and meaningful chart to show the experimental results. For all selected mycotoxins, is the established method analyzed simultaneously or one by one?

From the provided chromatogram, the retention time of the target is obviously overlapping, which is very difficult for accurate quantification. If the analysis of 8 target substances is performed simultaneously, a complete chromatogram is strictly required.

Please unify the units in the article. It will help the readers to learn the article.

In the manuscript, the use of abbreviations is very irregular and should be strictly in accordance with the requirements.

There are so many strategies for mycotoxins analysis, the author was suggested to offer one detailed comparison on the results, for example, a table.

This may be effective to show the merits of this research.

The conclusion part is recommended to be rewritten. It should be able to cover the research content and have summary characteristics. 

Author Response

Dear Reviewer 1

Please find attached the revised version of our paper entitled "QuEChERS LC-MS/MS screening method for mycotoxins detection in cereal products and spices". We have carefully revised our manuscript according to the reviewers' suggestions. We apologize for the delay, but we have done our best to respond to the requests of the reviewers in the best possible way, adhering as closely as possible to the high journal's standards. We have attached a word file with the track changes made to ease your perusal of our manuscript changes.

All the step by step changes are reported below:

  1. This research described a screening method for mycotoxin detection in cereal products and spices. Although the method might be effective for the selected mycotoxins, the obtained analysis results were not very advantageous, even if it was emphasized to be a QuEChERS method. It was not recommended for publication. There are so many strategies for mycotoxins analysis, the author was suggested to offer one detailed comparison on the results, for example, a table. This may be effective to show the merits of this research.
    • Dear reviewer, thank you for your opinion. We tried to do our best to validate this method. We know there are a lot of analytical techniques to analyze mycotoxins. However, as an "innovative approach" we use a semi-quantitative procedure according to (EC) No 401/2006. In this way, an LC-MS/MS can be used as screening, increasing the chance to discover new incidents with high exposure and health risks to consumers. Many methods are confirmatory, but they needed to be validated in each matrix. Our method can be validated in a single matrix for each commodity group and be compliant with EU Regulation. Furthermore, we have a numeric result even if with a screening method. This can be helpful to avoid the bottleneck on mycotoxins analysis and increase data on unique or difficult matrices.
    • Regarding the table, as you suggested, we would like to do it. Do you prefer that we compare the result of our method with screening methods or with confirmatory methods? Because we no have a LOD and a LOQ. Once again, thank you for your opinion.
  2. Figure 1 was unclear, please clarify them.
    • Dear reviewer, we changed Figure 1. Now it is Figure 3, and we hope it is ok now. (L426)
  3. Please unify the units in the article. It will help the readers to learn the article.
    • Dear reviewer, we unify the units as you suggested.
  4. In introduction part, it is recommended to provide more detailed information about the hazards of mycotoxins measured. The molecular structure of the mycotoxins tested should be provided, and more analysis of experimental results should be made accordingly.
    • Dear reviewer, we have made this change as suggested, and we added the info required. We added some molecular structure (Figure 1, L156) and more information about mycotoxins' hazards (L29-L88). We discussed more the results. (L273:L297).
  5. Although the author proposed that the established method is a QuEChERS method, the above characteristics are not seen from the results presented. Please add relevant research data such as detection time, accuracy, etc. in the manuscript.
    • Dear reviewer, we have made this change as suggested, and we added the info required (Table 4, L385).
  6. The chromatograms of the various substances provided are not clear, and the key information is missing. It is necessary to provide a complete and meaningful chart to show the experimental results. For all selected mycotoxins, is the established method analyzed simultaneously or one by one?
    • Dear reviewer, all mycotoxins are analyzed together with a single run. We added the total ion current (TIC) in Figure 3 (L425) and more data on Table 3 (L228) and Table 4 (L385).
  7. From the provided chromatogram, the retention time of the target is obviously overlapping, which is very difficult for accurate quantification. If the analysis of 8 target substances is performed simultaneously, a complete chromatogram is strictly required.
    • Dear reviewer, we added the TIC Figure 3 (L425).
  8. In the manuscript, the use of abbreviations is very irregular and should be strictly in accordance with the requirements.
    • Dear reviewer, we did as you suggested. We are sorry for the mistakes.
    •  
  9. The conclusion part is recommended to be rewritten. It should be able to cover the research content and have summary characteristics. 
    • Dear reviewer, we changed the conclusion part as recommended. (L465, L477).

Hope these changes could be helpful for the manuscript reconsideration.

Thank you for the privilege to consider our work in your esteemed journal.

King regards

Francesco Giuseppe Galluzzo

Reviewer 2 Report

Suggestions for introduction:

  • Some background on where the mycotoxins are introduced to the cereal products and spices.  Yes, they are fungus related, but some literature to describe why this is an issue to cereal and spices in particular would be helpful.
  • In lines 67-69, there is a very brief discussion of the use of QuEChERS.  Since this is the focus of the manuscript, this discussion should be expanded.
  • In section 2.1 (Chemical and standards), the QuEChERS appear to be a reagent, but it is really a tube filled with solid chemical.  It should be described separately from the reagents (methanol, etc.).
  • Line 135 lists r2.  It should be r2.
  • Equation 2 (below line 151) is misleading.  It looks like it could be ME = 100 x (1-Abm)/As.  The primary reference makes it clear the equation is ME = 100 x (1 - (Abm/As)).  Although it seems the authors intended the equation to reflect what was in the primary reference, the presentation could be improved to be clear to the reader.
  • In the first paragraph of section 3.1.1, the authors state that the best ratio of ACN/HCOOH was found to be 80:20.  The authors should include that other ratios were examined to reach this conclusion. 
  • In the second paragraph of section 3.1.1, the authors state that reference 60 describes 900 scientific papers using QuEChERS were published in the last 5 years.  However, reference 60 was published in 2015 so this is an incorrect statement.  Also, the authors write out 900 (nine hundred) but this value is >10 so it should be written numerically.
  • In the second paragraph of Section 3.1.2, the authors justify not using isotope-labeled internal standards. I agree with their justification.
  • In the final paragraph of section 3.2.2, the authors comment on the moderately overlapped peaks, which are shown in Figure 1.  The chromatograms in Figure 1 indicate that the overlapping peaks, in some cases, are more than moderately overlapping.  The authors need to explain their approach to determining peak areas for these overlapping peaks as this is extremely important in assessing the results.
  • Section 3.3.2 briefly describes the application of their method to 100 real samples.  The paragraph is very short with very little detail.  To convince the reader that their results provided evidence that the samples were compliant with (EC) No 1881/2006, more information needs to be provided.  Perhaps a table or some examples of the results could be added.  Otherwise, the brief paragraph does not convince the reader that the method really is applicable to real world samples.  Particularly with difference in matrices among the samples analyzed.  Also, more information on the collection of the samples would also be useful.  For example, were they collected from the same vendor or different vendors, etc.

Author Response

Dear Reviewer 2

Please find attached the revised version of our paper entitled “QuEChERS LC-MS/MS screening method for mycotoxins detection in cereal products and spices”. We have carefully revised our manuscript according to the reviewers’ suggestions. We apologize for the delay, but we have done our best to respond to the requests of the reviewers in the best possible way, adhering as closely as possible to the high journal's standards. We have attached a word file with the track changes made to ease your perusal of our manuscript changes.

All the step by step changes are reported below:

  1. Some background on where the mycotoxins are introduced to the cereal products and spices. Yes, they are fungus related, but some literature to describe why this is an issue to cereal and spices in particular would be helpful.
    • Dear reviewer, we added some info about mycotoxins and literature about the presence of mycotoxin in cereals and spices. Thank you for your suggestion. (L30:L88)
  2. In lines 67-69, there is a very brief discussion of the use of QuEChERS. Since this is the focus of the manuscript, this discussion should be expanded.
    • Dear reviewer, we added more info about QuEChERS extraction as you suggested (L126:144)
  3. In section 2.1 (Chemical and standards), the QuEChERS appear to be a reagent, but it is really a tube filled with solid chemical. It should be described separately from the reagents (methanol, etc.).
    • Dear reviewer, we added a new paragraph, and we described the tube separately. (L164)
  4. Line 135 lists r2. It should be r2.
    • Dear reviewer, we apologize for our grammar error. (L240)
  5. Equation 2 (below line 151) is misleading. It looks like it could be ME = 100 x (1-Abm)/As.  The primary reference makes it clear the equation is ME = 100 x (1 - (Abm/As)).  Although it seems the authors intended the equation to reflect what was in the primary reference, the presentation could be improved to be clear to the reader.
    • Dear reviewer, we improved the equation to be clearer to the reader (L256)
  6. In the first paragraph of section 3.1.1, the authors state that the best ratio of ACN/HCOOH was found to be 80:20. The authors should include that other ratios were examined to reach this conclusion.
    • Dear reviewer, we have made this change as suggested (L274) and we added a figure to compare extraction solvents (L298, Figure 2).
  7. In the second paragraph of section 3.1.1, the authors state that reference 60 describes 900 scientific papers using QuEChERS were published in the last 5 years. However, reference 60 was published in 2015 so this is an incorrect statement.  Also, the authors write out 900 (nine hundred) but this value is >10 so it should be written numerically.
    • Dear reviewer, we apologize for the mistake. We added “According to Web of Science, more than 1200 scientific papers covering aspects or use QuEChERS for extraction procedures.” (L128).
  8. In the second paragraph of Section 3.1.2, the authors justify not using isotope-labeled internal standards. I agree with their justification.
    • Dear reviewer, thank you for your understanding.
  9. In the final paragraph of section 3.2.2, the authors comment on the moderately overlapped peaks, which are shown in Figure 1. The chromatograms in Figure 1 indicate that the overlapping peaks, in some cases, are more than moderately overlapping.  The authors need to explain their approach to determining peak areas for these overlapping peaks as this is extremely important in assessing the results.
    • Dear reviewer, we adjusted Figure 3 (L428) to show better the retention time of analytes. We made mistakes with the figures. There is less overlapping between peaks.
  10. Section 3.3.2 briefly describes the application of their method to 100 real samples. The paragraph is very short with very little detail.  To convince the reader that their results provided evidence that the samples were compliant with (EC) No 1881/2006, more information needs to be provided.  Perhaps a table or some examples of the results could be added.  Otherwise, the brief paragraph does not convince the reader that the method really is applicable to real world samples.  Particularly with difference in matrices among the samples analyzed.  Also, more information on the collection of the samples would also be useful.  For example, were they collected from the same vendor or different vendors, etc.
    • Dear reviewer, we added Table 5 (L461) to explain better the results of real samples analysis. Samples were collected in an official inspection, and we cannot know more data on the origin of samples. We added all data we had. (L261:L267, L452:L458).

Hope these changes could be helpful for the manuscript reconsideration.

Thank you for the privilege to consider our work in your esteemed journal.

King regards

Francesco Giuseppe Galluzzo

Reviewer 3 Report

The article written by Licia Pantano et al. describes a new screening method for mycotoxins detection in cereal products and spices.

The material presented is valuable and has a novelty aspect. The proposed method of QuEChERS extraction is fast and cheap. This method can be used in the routine control of grain products and spices in the context of mycotoxins contained therein. It is worth noting that the authors indicate that their method was applied to screen mycotoxin in real 100 samples collected during an inspection in Sicily.

In my opinion, the article meets the requirements and can be published in the International Journal of Environmental Research and Public Health. To improve the reception of the article and to clarify some inaccurate sentences, I suggest applying the corrections below.

Line 59 "The are", should be "There are"?

Line 67 "(Quick Easy Cheap Effective Rugged e Safe)" should be (Quick Easy Cheap Effective Rugged and Safe)?

Line 77 QuEChERS is the name of the method, it can't be bought. The product 55227-U is Supel QuE Citrate (EN) Tube and 55228-U is Supel QuE PSA (EN) Tube. The description should be changed.

Line 97 MeOH/H2O (50/50) - is it v/v ratio?

Line 106 - subscript in H2O

Line 159 is it v/v ratio?

Fig 1 and the caption should be corrected. The caption "mycotoxin peaks" should be modified. A low-quality figure, not legible, it is worth inserting illustrations with higher resolution and reformatting the entire figure to take up less space and allow compare all peaks. 

Author Response

Dear Reviewer 3

Please find attached the revised version of our paper entitled “QuEChERS LC-MS/MS screening method for mycotoxins detection in cereal products and spices”. We have carefully revised our manuscript according to the reviewers’ suggestions. We apologize for the delay, but we have done our best to respond to the requests of the reviewers in the best possible way, adhering as closely as possible to the high journal's standards. We have attached a word file with the track changes made to ease your perusal of our manuscript changes.

All the step by step changes are reported below:

  1. Line 59 "The are", should be "There are"?
    • Dear reviewer, we apologize for the grammar mistake. (L117)
  2. Line 67 "(Quick Easy Cheap Effective Rugged e Safe)" should be (Quick Easy Cheap Effective Rugged and Safe)?
    • Dear reviewer, we apologize (once again) for the grammar mistake. (L126)
  3. Line 77 QuEChERS is the name of the method, it can't be bought. The product 55227-U is Supel QuE Citrate (EN) Tube and 55228-U is Supel QuE PSA (EN) Tube. The description should be changed.
    • Dear reviewer, we changed the description as you suggested. We moved the product into a different paragraph. (L164)
  4. Line 97 MeOH/H2O (50/50) - is it v/v ratio?
    • Dear reviewer, we added the “v/v” ratio. (L196)
  5. Line 106 - subscript in H2O
    • Dear reviewer, we apologize for the mistake. Thank you for the suggestion. We use “water” now. (L208).
  6. Line 159 is it v/v ratio?
    • Dear reviewer, we specify. (L273)
  7. Fig 1 and the caption should be corrected. The caption "mycotoxin peaks" should be modified. A low-quality figure, not legible, it is worth inserting illustrations with higher resolution and reformatting the entire figure to take up less space and allow compare all peaks. 
    • Dear reviewer, thank you for your precious suggestion. We tried to improve the quality of figures and tried to take up less space. (Figure 3, L428).

Hope these changes could be helpful for the manuscript reconsideration.

Thank you for the privilege to consider our work in your esteemed journal.

King regards

Francesco Giuseppe Galluzzo

Reviewer 4 Report

Major comments.

Since the commercial kits 55227-U and 55228-U from SIGMA ALDRICH are used in the sample preparation step It is necessary to better emphasize what is the innovation in the present manuscript. Furthermore, such kits are mentioned on line 77 only while in the section 2.3 “Sample extraction” (line 95) it is mentioned instead the solid phase Supelclean PSA from SIGMA ALDRICH (cod. 52738-U) never mentioned before. Clarify.

Line 88, sample extraction. Authors say: “About 5.0 ± 0.1 g of the sample was weighted in a falcon tube…”. The question is: were the corn and coffee beans ground or not? If not why? If yes, by means of which homogenizer? At what speed and temperature?

There is a question the authors should answer. In the manuscript it is clearly said that “The ionization in positive mode with the ammonium adduct was successful for all analytes” (lines 236-237). So, why in Table 3 the parent ion for OTA is 402.2 m/z? Being 404 amu the molecular weight of OTA, the parent ion of the ammonium adduct should be 422 m/z. Similarly, the parent ion of ZEA should be 336 m/z and so on.

Lines 237-238. Authors say: “The chromatographic run resolved all mycotoxin with an adequate resolution and in a run of six minutes (Figure 1)”. First of all, the sentence is not clear to the reader because all peaks (Figure 1) fall within a restricted area (2.85, 2.81, 2.76, 2.77, 1.93, 3.41, 3.16, 3.30, 3.52, 3.40 min) and it could seem that there are a number of coalescences. I propose a clearer sentence such like this: “All mycotoxins were detected within 4 minutes. Adequate resolution was obtained between ions with the same m/z which would be indistinguishable from the mass spectrometer if they coalesced”. Second: from Figure 1 AFLA-G1 and AFLA-G2 seem to have the same m/z (243.055) and the same RT (2.76 and 2.77 min). How is it possible to distinguish them in a real sample with the proposed method? Probably there is a mistake in Figure 1 or in the experimental conditions since Table 3 correctly reports that the daughter ions for AFLA-G1 and AFLA-G2 differ by 2 amu.

Lines 192-193. Authors say: “Isotope-labeled internal standards are very expensive and do not exist for all mycotoxins, and recovery is not required in the screening method”. Nevertheless, in Table 1 it is listed an isotope-labeled internal standard (OTA-d5) and it is not clear how and where it is used in the work.

Section 3.3.1, Ring test. The proficiency test GC+C2053 has an assigned value of 7530 ppb for OTA (7.53 mg/kg), while the reported result for OTA in maize is 2.53 ppb and is 1.85 ppb for OTA in black pepper (section 3.3.2). The proficiency test is out range of too many orders of magnitude to validate the levels measured in the real samples. The same can be said for the proficiency test MA2050.

Other comments.

In general, the work is not well organized. There are scattered errors and repetitions. Some parts are unclear or they are where they don't need to be. English language in some cases needs improvement. Below is a list of these issues.

Line 94. Here and throughout the manuscript: do not confuse supernatant with surfactant

Line 208. Replace “matrixes” with “matrices

Line 120: “4800 kV”. It is likely this value is wrong, probably the correct voltage is 4.8 kV

Lines 70-71: “…a fast and reliable LC-MS/MS method followed by QuEChERS extraction”. The extraction comes before the LC-MS/MS. I suggest: “…a QuEChERS extraction together with a fast and reliable LC-MS/MS method

Table 1. In the last two columns the unit of concentration lacks.

Line 119: “Sheath gas pressure 40, auxiliary gas pressure 30”. Units lack, “psi” I think.

Lines 84-85. SMix1 and SMix2 are mentioned here only. Explain their use and how they were made

Line 59: “The are several analytical techniques” change to “There are several analytical techniques”

Lines 121-122: “…was used to record and elaborated data”. Replace with “…was used to record and elaborate data

Line 136. Replace “analyze” with “by analyzing”.

Line 141. Replace “that change” with “of changing”.

Line 95: “After one minute of handle shacking, centrifugated for five minutes”. Replace with “After one minute of handle shacking and five minutes of centrifugation”. At the end of the sentence it is advisable to add “the sample was ready for the injection

Line 192: “However, internal standards are very expensive”. Replace with: “However, isotope-labeled internal standards are very expensive

The section “samples” lacks. Lines 281-282 should be moved in Materials and Methods and expanded

Line 61 and line 66. Wrong references: [10,41-48] delete 10; [41,41,48] 41 repeated

Line 42. Instead of “It was proven that 25% of cereals approximately consumed in the world are contaminated by mycotoxins [30]” it is better to write: “It was proven that 25% of cereals approximately consumed in the world in 1998 were contaminated by mycotoxins [30]” because the reference [30] reports such sentence by referring to another work of 1998. Otherwise it could seem that 25% refers to today

Lines 122-123: “Results obtained from the direct infusion of the analyte can be seen in Table 3”. Were retention times in Table 3 obtained from the direct infusion?

The section 3.1.1 is too long and too dispersive it should be summarized and moved to Introduction

Lines 196-199 should be moved to Introduction

Lines 210-214. Not clear: not all these matrices seem to have been analyzed in the present work

Line 200: “There is a strong ME…”. It should be better “In the present work a strong ME was observed...”

Lines 145-147. It is not clear the linearity range, if the calibration is external, if it is a matrix-matched calibration, if it is standard addition method, there seems to be no LOD declared, no LOQ declared, no blanks, no recoveries. What means “semi-quantitative”? Explain in detail

Author Response

Dear Reviewer 4

Please find attached the revised version of our paper entitled “QuEChERS LC-MS/MS screening method for mycotoxins detection in cereal products and spices”. We have carefully revised our manuscript according to the reviewers’ suggestions. We apologize for the delay, but we have done our best to respond to the requests of the reviewers in the best possible way, adhering as closely as possible to the high journal's standards. We have attached a word file with the track changes made to ease your perusal of our manuscript changes.

All the step by step changes are reported below:

  1. Since the commercial kits 55227-U and 55228-U from SIGMA ALDRICH are used in the sample preparation step It is necessary to better emphasize what is the innovation in the present manuscript. Furthermore, such kits are mentioned on line 77 only while in the section 2.3 “Sample extraction” (line 95) it is mentioned instead the solid phase Supelclean PSA from SIGMA ALDRICH (cod. 52738-U) never mentioned before. Clarify.
    • Dear reviewer, the composition of 55228-U is “magnesium sulfate, 900 mg Supelclean™ PSA, and 150 mg (Cat. No. 52738-U)” and therefore, we mentioned only 55228-U because it contains Cat. No. 52738-U (L164:L168).
  2. Line 88, sample extraction. Authors say: “About 5.0 ± 0.1 g of the sample was weighted in a falcon tube…”. The question is: were the corn and coffee beans ground or not? If not why? If yes, by means of which homogenizer? At what speed and temperature?
    • Dear reviewer, thank you for your suggestion. We added more info as you required. (L177:L182).
  3. There is a question the authors should answer. In the manuscript it is clearly said that “The ionization in positive mode with the ammonium adduct was successful for all analytes” (lines 236-237). So, why in Table 3 the parent ion for OTA is 402.2 m/z? Being 404 amu the molecular weight of OTA, the parent ion of the ammonium adduct should be 422 m/z. Similarly, the parent ion of ZEA should be 336 m/z and so on.
    • Dear reviewer, we apology for the mistakes. Thank you for your precious suggestion. We corrected the mistakes. (Table 3, L228).
  4. Lines 237-238. Authors say: “The chromatographic run resolved all mycotoxin with an adequate resolution and in a run of six minutes (Figure 1)”. First of all, the sentence is not clear to the reader because all peaks (Figure 1) fall within a restricted area (2.85, 2.81, 2.76, 2.77, 1.93, 3.41, 3.16, 3.30, 3.52, 3.40 min) and it could seem that there are a number of coalescences. I propose a clearer sentence such like this: “All mycotoxins were detected within 4 minutes. Adequate resolution was obtained between ions with the same m/z which would be indistinguishable from the mass spectrometer if they coalesced”. Second: from Figure 1 AFLA-G1 and AFLA-G2 seem to have the same m/z (243.055) and the same RT (2.76 and 2.77 min). How is it possible to distinguish them in a real sample with the proposed method? Probably there is a mistake in Figure 1 or in the experimental conditions since Table 3 correctly reports that the daughter ions for AFLA-G1 and AFLA-G2 differ by 2 amu.
    • Dear reviewer, we change the line as you suggested (L394-396). Moreover, thank you again for your opinion; we apologize for the mistakes we made in Figure 1. We corrected it. (L428, Figure 3).

  1. Lines 192-193. Authors say: “Isotope-labeled internal standards are very expensive and do not exist for all mycotoxins, and recovery is not required in the screening method”. Nevertheless, in Table 1 it is listed an isotope-labeled internal standard (OTA-d5) and it is not clear how and where it is used in the work.
    • Dear reviewer, we added in the text how we use OTA-d5 (L-184, L345:L346).
  2. Section 3.3.1, Ring test. The proficiency test GC+C2053 has an assigned value of 7530 ppb for OTA (7.53 mg/kg), while the reported result for OTA in maize is 2.53 ppb and is 1.85 ppb for OTA in black pepper (section 3.3.2). The proficiency test is out range of too many orders of magnitude to validate the levels measured in the real samples. The same can be said for the proficiency test MA2050.
    • Dear reviewer, we are sorry. These were typos. The font should be “Symbol” (m) and not Palatino Linotype (m). (L437)

Other comments.

  1. Line 94. Here and throughout the manuscript: do not confuse supernatant with surfactant
    • Dear reviewer, thank you for the advice. We corrected the terms.
  2. Line 208. Replace “matrixes” with “matrices”
    • Dear reviewer, we are sorry for the grammar error. (L361)
  3. Line 120: “4800 kV”. It is likely this value is wrong, probably the correct voltage is 4.8 Kv
    • Dear reviewer, we have made this change as suggested (L222)
  4. Lines 70-71: “…a fast and reliable LC-MS/MS method followed by QuEChERS extraction”. The extraction comes before the LC-MS/MS. I suggest: “…a QuEChERS extraction together with a fast and reliable LC-MS/MS method”
    • Dear reviewer, we have made this change as suggested (L148)
  5. Table 1. In the last two columns the unit of concentration lacks.
    • Dear reviewer, we add the unit of concentration in the caption (L198)
  6. Line 119: “Sheath gas pressure 40, auxiliary gas pressure 30”. Units lack, “psi” I think.
    • Dear reviewer, we add the units. (L219)
  7. Lines 84-85. SMix1 and SMix2 are mentioned here only. Explain their use and how they were made
    • Dear reviewer, we explain the use and how they were made in Table 1 (L174, L198)
  8. Line 59: “The are several analytical techniques” change to “There are several analytical techniques”
    • Dear reviewer, we changed the line as you suggested. (L117).
  9. Lines 121-122: “…was used to record and elaborated data”. Replace with “…was used to record and elaborate data”
    • Dear reviewer, we replaced the line as you suggested. (L223).
  10. Line 136. Replace “analyze” with “by analyzing”.
    • Dear reviewer, we are sorry for the grammar mistake. (L240).
    •  
  11. Line 141. Replace “that change” with “of changing”.
    • Dear reviewer, we are sorry for the grammar mistake. (L245)
  12. Line 95: “After one minute of handle shacking, centrifugated for five minutes”. Replace with “After one minute of handle shacking and five minutes of centrifugation”. At the end of the sentence it is advisable to add “the sample was ready for the injection”
    • Dear reviewer, we changed the lines as you suggested (L195, L197)
  13. Line 192: “However, internal standards are very expensive”. Replace with: “However, isotope-labeled internal standards are very expensive”
    • Dear reviewer, we changed the lines as you suggested (L341)
  14. The section “samples” lacks. Lines 281-282 should be moved in Materials and Methods and expanded
    • Dear reviewer, we added some info and we moved “samples” in Materials and Methods. (L260:L266)
  15. Line 61 and line 66. Wrong references: [10,41-48] delete 10; [41,41,48] 41 repeated
    • Dear reviewer, we changed the references as you suggested.
  16. Line 42. Instead of “It was proven that 25% of cereals approximately consumed in the world are contaminated by mycotoxins [30]” it is better to write: “It was proven that 25% of cereals approximately consumed in the world in 1998 were contaminated by mycotoxins [30]” because the reference [30] reports such sentence by referring to another work of 1998. Otherwise it could seem that 25% refers to today
    • Dear reviewer, we apologize for mistake. We changed. (L97:L98)
  17. Lines 122-123: “Results obtained from the direct infusion of the analyte can be seen in Table 3”. Were retention times in Table 3 obtained from the direct infusion?
    • Dear reviewer, we apologize again for the mistakes. We clarified it. (L224).
  18. The section 3.1.1 is too long and too dispersive it should be summarized and moved to Introduction
    • Dear reviewer, we reduced text and we moved the lines in the introduction.
  19. Lines 196-199 should be moved to Introduction
    • Dear reviewer, we deleted the lines.
  20. Lines 210-214. Not clear: not all these matrices seem to have been analyzed in the present work
    • Dear reviewer, according to (EC) No 401/2006 the screening method can validate a single matrix for each commodity groups. In this case, we validated wheat and black pepper that can be used for the mycotoxin analysis of the following commodity groups: “High starch and/or protein content and low water and fat content” and “Difficult or unique commodities”. Thanks to your precious suggestion, we added few lines to specify better matrices validated (L362, L368).
  21. Line 200: “There is a strong ME…”. It should be better “In the present work a strong ME was observed...”
    • Dear reviewer, we changed the lines as you suggested (L350:L351)
  22. Lines 145-147. It is not clear the linearity range, if the calibration is external, if it is a matrix-matched calibration, if it is standard addition method, there seems to be no LOD declared, no LOQ declared, no blanks, no recoveries. What means “semi-quantitative”? Explain in detail
    • Dear reviewer, the linearity range was obtained with external calibration. However, the linearity range is different for each mycotoxin. Semi-quantitative methods are screening method that are allowed by (EC) No 401/2006 that express a number as results. They can be physicochemical methods based on chromatography or direct detection by mass spectrometry.
    • Screening target concentration (STC) is the concentration of interest for detection of the mycotoxin in a sample that is equal to the applicable maximum level (if present). For screening methods, a Cut-off level is required for validation.
    • If a sample results with a mycotoxin content above cut off level, it triggers a confirmatory test.
    • For this reason, we no have LOQ and LOD but Cut-Off levels.
    • We added recoveries (Table 4, L385) and we analyzed blanks in validation procedure to obtain the corresponding rate of false suspect results.
    •  

Hope these changes could be helpful for the manuscript reconsideration.

Thank you for the privilege to consider our work in your esteemed journal.

King regards

Francesco Giuseppe Galluzzo

Round 2

Reviewer 4 Report

I am grateful to the authors for promptly answering to all questions raised, for clarifying some points and for correcting the text according to the suggested changes.

Unfortunately they did not adequately answer to one of the principal questions which again is reported below.

<>.

Authors responded by substituting in Table 3 the value 402.2 m/z, for OTA, with the value 404.2 m/z.

The fact is when ammonium adducts are used the species [M+NH4]+ forms, where M is the molecular weight of the analyte while the molecular weight of NH4+ is 18. The MW of OTA is 404 amu, therefore the OTA-adduct (parent ion) will be 404+18 = 422 m/z. The same is valid for ZEA (318.4+18=336.4), AFB1 (312.3+18=330.3), AFB2 (314.3+18=332.3), AFG1 (328.3+18=346.3), AFG2 (330.3+18=348.3) and so on. In Table 3 all parent ions (with the exception of T2 and HT2) erroneously report the MW of the analyte without the adduct. Why? It cannot be a distraction because the anomaly was already reported by us in the first review cycle. Did authors observe in the mass spectrum a fragment of 404 m/z for OTA-adduct? Or did they observe the fragment 422 m/z? In the first case the ammonium adduct was not formed, as they declare in the manuscript. But in such case they should have replied to me in the first cycle of review that the adduct had not formed. By substituting 402.2 m/z with 404.2 m/z and leaving unmodified the other precursor ions authors have shown that they are not familiar with the LC-MS/MS technique which should be the main part of this work. The following paper can be useful in this context, doi: 10.1093/jat/bkw002.

I am sorry, in my opinion the manuscript is not suitable for publication and I hope these observations will be useful for your future works for which I encourage you to continue as the subject is of interest being related to food safety. 

Author Response

Dear Reviewer 4

Please find attached the revised version of our paper entitled “QuEChERS LC-MS/MS screening method for mycotoxins detection in cereal products and spices”. We have carefully revised our manuscript according to the reviewers’ suggestions. We apologize for the delay, but we have done our best to respond to the requests of the reviewers in the best possible way, adhering as closely as possible to the high journal's standards. We have attached a word file with the track changes made to ease your perusal of our manuscript changes.

All the step by step changes are reported below:

I am grateful to the authors for promptly answering to all questions raised, for clarifying some points and for correcting the text according to the suggested changes. Unfortunately they did not adequately answer to one of the principal questions which again is reported below.

<<In the manuscript it is clearly said that “The ionization in positive mode with the ammonium adduct was successful for all analytes” (lines 236-237). So, why in Table 3 the parent ion for OTA is 402.2 m/z? Being 404 amu the molecular weight of OTA, the parent ion of the ammonium adduct should be 422 m/z. Similarly, the parent ion of ZEA should be 336 m/z and so on>>.

Authors responded by substituting in Table 3 the value 402.2 m/z, for OTA, with the value 404.2 m/z.

The fact is when ammonium adducts are used the species [M+NH4]+ forms, where M is the molecular weight of the analyte while the molecular weight of NH4+ is 18. The MW of OTA is 404 amu, therefore the OTA-adduct (parent ion) will be 404+18 = 422 m/z. The same is valid for ZEA (318.4+18=336.4), AFB1 (312.3+18=330.3), AFB2 (314.3+18=332.3), AFG1 (328.3+18=346.3), AFG2 (330.3+18=348.3) and so on. In Table 3 all parent ions (with the exception of T2 and HT2) erroneously report the MW of the analyte without the adduct. Why? It cannot be a distraction because the anomaly was already reported by us in the first review cycle. Did authors observe in the mass spectrum a fragment of 404 m/z for OTA-adduct? Or did they observe the fragment 422 m/z? In the first case the ammonium adduct was not formed, as they declare in the manuscript. But in such case they should have replied to me in the first cycle of review that the adduct had not formed. By substituting 402.2 m/z with 404.2 m/z and leaving unmodified the other precursor ions authors have shown that they are not familiar with the LC-MS/MS technique which should be the main part of this work. The following paper can be useful in this context, doi: 10.1093/jat/bkw002.

I am sorry, in my opinion the manuscript is not suitable for publication and I hope these observations will be useful for your future works for which I encourage you to continue as the subject is of interest being related to food safety. 

  • Dear reviewer, thank you for your opinion and for your accurate revision. We are sorry for all these mistakes about ions. For OTA, we corrected the parent ion because it was a typo. We deleted the lines “ The ionization in positive mode with the ammonium adduct was successful for all analytes” (L 395-396) ” in the first revision because the ammonium adducts were formed only for T2 and HT2. However, was our mistake to not refer the deleted lines in the first response. For other mycotoxins, the protonated ion was used as a precursor. We adjusted Table 3 (L227) and in the text (L 393-395).
  • Thanks for your useful observations and for the time spent in this revision.

Hope these changes could be helpful for the manuscript reconsideration.

Thank you for the privilege to consider our work in your esteemed journal.

King regards

Francesco Giuseppe Galluzzo
